# NKG2C+ NK Cells for Immunotherapy of Glioblastoma Multiforme

**DOI:** 10.3390/ijms23105857

**Published:** 2022-05-23

**Authors:** Shafiq Murad, Susanne Michen, Alexander Becker, Monika Füssel, Gabriele Schackert, Torsten Tonn, Frank Momburg, Achim Temme

**Affiliations:** 1Department of Neurosurgery, Section Experimental Neurosurgery/Tumor Immunology, University Hospital Carl Gustav Carus, TU Dresden, Fetscherstr. 74, 01307 Dresden, Germany; shafiq.murad@ukdd.de (S.M.); susanne.michen@ukdd.de (S.M.); alexander.becker@ukdd.de (A.B.); gabriele.schackert@ukdd.de (G.S.); 2DKMS Life Science Lab GmbH, St. Petersburger Str. 2, 01069 Dresden, Germany; fuessel@dkms-lab.de; 3German Cancer Consortium (DKTK), Partner Site Dresden and German Cancer Research Center (DKFZ), 69120 Heidelberg, Germany; 4National Center for Tumor Diseases (NCT/UCC), 01307 Dresden, Germany; 5Department of Transfusion Medicine, Medical Faculty Carl Gustav Carus, Technische Universität Dresden, Fetscherstr. 74, 01307 Dresden, Germany; t.tonn@blutspende.de; 6Antigen Presentation and T/NK Cell Activation Group (D121), German Cancer Research Center (DKFZ), Im Neuenheimer Feld 280, 69120 Heidelberg, Germany; f.momburg@dkfz.de; 7Clinical Cooperation Unit “Applied Tumor Immunity” (D120), German Cancer Research Center (DKFZ), Im Neuenheimer Feld 280, 69120 Heidelberg, Germany

**Keywords:** brain cancer, glioblastoma, HLA-E, HLA-G, NK cells, immunotherapy

## Abstract

In glioblastoma, non-classical human leucocyte antigen E (HLA-E) and HLA-G are frequently overexpressed. HLA-E loaded with peptides derived from HLA class I and from HLA-G contributes to inhibition of natural killer (NK) cells with expression of the inhibitory receptor CD94/NKG2A. We investigated whether NK cells expressing the activating CD94/NKG2C receptor counterpart were able to exert anti-glioma effects. NKG2C+ subsets were preferentially expanded by a feeder cell line engineered to express an artificial disulfide-stabilized trimeric HLA-E ligand (HLA-E*spG). NK cells expanded by a feeder cell line, which facilitates outgrowth of conventional NKG2A+, and fresh NK cells, were included for comparison. Expansion via the HLA-E*spG feeder cells selectively increased the fraction of NKG2C+ NK cells, which displayed a higher frequency of KIR2DL2/L3/S2 and CD16 when compared to expanded NKG2A+ NK cells. NKG2C+ NK cells exhibited increased cytotoxicity against K562 and KIR:HLA-matched and -mismatched primary glioblastoma multiforme (GBM) cells when compared to NKG2A+ NK cells and corresponding fresh NK cells. Cytotoxic responses of NKG2C+ NK cells were even more pronounced when utilizing target cells engineered with HLA-E*spG. These findings support the notion that NKG2C+ NK cells have potential therapeutic value for treating gliomas.

## 1. Introduction

Despite rapid progress in the treatment of many cancers, glioblastoma multiforme (GBM) remains a devastating disease with dismal prognosis [1]. The multimodal standard treatment of GBM comprises surgical resection, followed by radio-chemotherapy plus temozolomide and temozolomide maintenance [2,3]. Nevertheless, the median survival of patients remains 14.6 months [2], which emphasizes the need for developing novel and more efficient adjuvant treatments.

Specifically, as a treatment strategy for human solid cancers, a plethora of promising immunotherapies have entered the stage. However, so far, vaccination strategies [4] and immune checkpoint blockade [5,6] has been hampered by the immune suppressive tumor microenvironment (i.e., regulatory T cells and myeloid-derived suppressor cells, MDSCs) and low mutational load of GBM [7] resulting in disappointing clinical responses (for reviews see [8,9]). On the other hand, cellular immunotherapy using chimeric antigen receptor-modified (CAR) T cells targeting the tumor-associated antigens EGFRvIII and IL-13R2α on the surface of GBM cells showed encouraging clinical responses [10,11].

Although most studies have focused on T cells for adoptive immunotherapy of GBM, few have investigated the potential of natural killer (NK) cells. NK cells develop from CD34+ hematopoietic progenitors, are characterized by CD56+ CD3− surface expression and comprise about 5–15% of circulating lymphocytes. They belong to the innate immune system but possess the same cytotoxic weapons (i.e., perforin/granzyme B, death receptor ligands) as cytotoxic T lymphocytes. NK cells can be transplanted across human leucocyte antigen (HLA)-boundaries and because of their shorter life span and their inability to produce interleukin-2 (IL-2), massive off-target side effects (i.e., brain edema) can be limited when compared to tumor-specific T cells as well as CAR-engineered T cells [12]. In contrast to T cells, NK cells have the intrinsic capacity to destroy virus-infected and malignant cells. This is achieved by a set of endogenous receptors sensing self and activating ligands on target cells, respectively. More specifically, NK cells express in a variegated pattern inhibitory and activating receptors, such as the polymorphic receptors of the killer cell immunoglobulin-like receptor (KIR) family, natural cytotoxicity receptors (NCRs) (i.e., NKp30, NKp40, NKp44), and members of the C-type lectin receptors such as NKG2D recognizing stress ligands MICA and ULBPs on target cells [13]. The cytotoxic response of NK cells is tuned by the incoming net sum of inhibitory and activating signals, and under physiological conditions NK cells are unresponsive to “self” [14].

In humans, there are two dominant inhibitory NK cell receptor types recognizing HLA molecules and thereby providing tolerance to self: the C-type lectin NKG2A, which assembles with its co-receptor CD94, and inhibitory KIRs (iKIRs). Whereas iKIRs directly bind to cognate HLA class I alleles, CD94/NKG2A binds to non-classical HLA-E presenting HLA class I leader peptides [15]. Thereby CD94/NKG2A indirectly senses the presence of HLA class I on cells. The polymorphic KIR family also comprises activating KIRs (aKIRs), also designated “short-tailed” killer Ig–like receptors, which signal through immunoreceptor tyrosine-based activation motif (ITAM)-containing signaling adapter DAP12 [16]. The aKIRs arose from gene duplication and conversion from iKIRs, have lost their intracellular protein domains and bind with lower affinity to HLA class I molecules than their inhibitory counterparts [17,18].

There is accumulating evidence that deregulated expression of HLA-E in GBM contributes to inhibition of conventional NKG2A+ NK cells [19,20]. In addition to peptides derived from classical HLA, the immune checkpoint-related non-classical HLA-G, frequently expressed in GBM [21], provides the nonameric peptide VMAPRTLFL, which when loaded on HLA-E results in HLA-E:peptide complexes reported to have the best affinity for inhibiting CD94/NKG2A and activating CD94/NKG2C receptors [22,23].

The activating CD94/NKG2C receptor, the counterpart of CD94/NKG2A, is devoid of an intracellular domain containing an immunoreceptor tyrosine-based inhibitory motif (ITIM) and instead assembles with the signaling adapter DAP12 [24]. Interestingly, signals via DAP12 transmit sufficient activating signals to overcome protective levels of HLA class I on solid tumor cells, including glioma cells, as previously shown by our group when using a DAP12-based CAR [25,26]. Of note, a higher frequency of terminally differentiated CD94/NKG2C+ NK cell subsets (referred to as NKG2C+ NK cells), devoid of NKG2A, phenotypically skewed to express KIRs and positive for the terminal differentiation marker CD57, are preferentially found in human cytomegalovirus (HCMV)-seropositive individuals [27]. Many reports demonstrated that HCMV infection and concomitant loading of HCMV-UL40 derived nonameric peptides, which share homology to endogenous peptides derived from classical HLA molecules and HLA-G, on HLA-E represent powerful stimuli to promote outgrowth and a cytotoxic response of NKG2C+ NK cells [28,29]. Furthermore, the NKG2C+ NK cell subset exhibits broad epigenetic modifications, expands during reactivation of HCMV and is transplantable, which are hallmarks of adaptive cells (for reviewing see [30]). We therefore concluded that immunotherapy using NKG2C+ NK cells might be a reasonable approach to treat GBM. Yet, one bottleneck for clinical application of NKG2C+ NK cells is the often-low abundance of NKG2C+ NK cells in peripheral blood. In this study, we employed a selective expansion of NKG2C+ NK cells using a feeder cell line with ectopic expression of IL-2, membrane bound IL-15, and further modified with an artificial disulfide-stabilized trimeric HLA-E-ligand (HLA-E*spG). NK cells expanded by HLA-E*spG-modified feeder cells showed a selective expansion of NKG2C+ NK cells. These cells displayed increased surface expression of KIR2DL2/L3/S2 and CD16 when compared to NKG2A+ NK cells, which preferentially expands on parental feeder cells devoid of HLA-E*spG (referred to as NKG2A+ NK cells). NKG2C+ NK cells exhibited natural cytotoxicity against HLA-ABC+/HLA-E+/HLA-G+ primary glioblastoma cells which was increased by KIR:HLA mismatch, whereas corresponding fresh NK cells from peripheral blood and NKG2A+ NK cells showed no cytotoxic response. Moreover, ectopic overexpression of HLA-E*spG on target cells further enhanced the cytotoxic response of NKG2C+ NK cells, demonstrating that strong HLA-E:NKG2C ligation can contribute to recognition and killing of glioblastoma cells by NK cells.

## 2. Results

### 2.1. Expansion of NKG2C+ NK Cells Using Feeder Cells

Exploitation of NKG2C+ NK cells for immunotherapy of GBM cells is hampered by low abundance of NK cells in peripheral blood of donors when compared to T cells and furthermore, can be impeded by often inadequate frequencies of NKG2C+ cells in HCMV-seropositive donors. When testing 39 HCMV-seropositive donors, we found only four donors with NGK2C frequencies ranging from 35% to 75%. However, the mean frequency of NKG2C+ NK cells among CD56+/CD3− cells was only 12.3% (±9.3%), which was, as expected from the literature, higher when compared to healthy donors (NKG2C frequency 4.6% ± 2.3%) (Appendix A) [27]. Therefore, we sought to expand NKG2C+ NK cells from peripheral blood for investigating their potential cytotoxicity towards GBM target cells. Recently, our group developed an HLA class I-positive NK feeder cell line originating from the adherent prostate carcinoma cell line PC3 and genetically engineered to express IL-2, 4-1BBL and a membrane-bound form of IL-15 (PC3^PSCA^-IL-2-4-1BBL-mIL-15d). This feeder cell line enabled efficient outgrowth of NK cells from peripheral blood mononuclear cells (PBMCs) and expansion of purified NK cells without supplementing exogenous cytokines [31]. Of note, PC3 cells and this feeder cell line are characterized by robust expression levels of endogenous HLA-E and HLA-G, the latter providing a high affinity leader peptide for HLA-E. Tentatively, this phenotypic setting should favor the outgrowth of NKG2C+ NK cells, but in the previous study only modest numbers of NKG2C-positive NK cells emerged during cultivation with PC3^PSCA^-IL-2-4-1BBL-mIL-15d feeder cells [31]. Yet, all donors included in this study were tested seronegative for HCMV. In this regard, it was deemed conceivable that this feeder cell line might support the preferential outgrowth of NKG2C+ NK cells from HCMV-seropositive donors. When expanding NK cells from HCMV-seropositive donors with PC3^PSCA^-IL-2-4-1BBL-mIL-15d feeder cells we again monitored outgrowth of the NKG2C+ NK cell subset, from 6.9% (mean ± 7.8%) observed in fresh NK cells to 18.7% (mean ± 10.7%) in expanded NK cells (Appendix A). This result was better when compared to previous results obtained with NK cells from HCMV-seronegative donors [31]. Yet, frequency and absolute numbers of NKG2C+ NK cells were still considered inappropriate for the intended study.

To improve selective expansion of NKG2C+ NK cells, a new feeder cell line based on previously described PC3^PSCA^-IL-2-mIL-15d feeder cells [31] was generated by genetic modification with an artificial disulfide-stabilized single-chain trimer (dSCT) consisting of β2-microglobulin and the HLA-G signal peptide derived nonamer VMAPRTLFL fused to HLA-E (HLA-E*spG) [22]. As recently described, such recombinant HLA-E*spG, when coupled to beads, enabled crosslinking of NKG2C receptors and induced improved activation and cytokine release of NKG2C+ NK cells when compared to dSCTs presenting HSP60- or HLA-class I-derived peptides. We chose PC3^PSCA^-IL-2-mIL-15d feeder cells for modification with HLA-E*spG since this cell line gives only suboptimal support for NK cells and considered that the endogenous expression of classical HLA molecules, HLA-E and HLA-G in combination with ectopic expression of IL-2, membrane-bound IL-15 and HLA-E*spG could improve a selective expansion of NKG2C+ NK cells. The new feeder cell line, designated PC3^PSCA^-IL-2-mIL-15d-HLA-E*spG, was generated as described in the material and methods section by transduction with a lentiviral vector encoding HLA-E*spG (Figure 1a). After antibiotic selection, the expression of the HLA-E*spG transgene was confirmed by HLA-E surface staining and flow cytometry analysis. More specifically, the expression of HLA-E*spG resulted in a ten-fold increased mean fluorescence intensity (MFI) for HLA-E signals when compared to parental PC3^PSCA^-IL-2-mIL-15d feeder cells (Figure 1c).

In subsequent experiments NK cells were expanded by co-cultivation with both PC3^PSCA^-IL-2-mIL-15d-HLA-E*spG and PC3^PSCA^-IL-2-mIL-15d feeder cells (Figure 2a). In general, NK cells from donors displaying CD56+/NKG2C+ NK cell frequencies below 4% did not enable significant expansion of NKG2C+ NK cells (data not shown) and therefore were excluded from further analysis. So far, co-cultivation of purified NK cells from six donors with PC3^PSCA^-IL-2-mIL-15d-HLA-E*spG feeder cells resulted at day 14 in more than 8.8-fold (± 1.7) increased total cell numbers, which was similar when expanding with PC3^PSCA^-IL-2-mIL-15d feeder cells (6.8 ± 1.1) (Figure 2b). Furthermore, initial mean frequencies of NK cells displaying surface NKG2C receptors before expansion was 8.9% (mean ± 7%) and after 14 days of expansion 54.0% (mean ± 20.9%). More specifically, the HLA-E*spG-expressing feeder cell line was capable of inducing significant outgrowth of NK cells single-positive for NKG2C when compared to single-positive NKG2C frequencies obtained in fresh NK cells as well as NK cells expanded with PC3^PSCA^-IL-2-mIL-15d feeder cells (Figure 2c). However, a higher donor dependent variance was observed, which in some cases resulted in smaller relative amounts of NKG2C single-positive NK cells which was accompanied by the appearance of NK cells double-positive for NKG2A and NKG2C and remaining NKG2A single-positive cell fractions (Figure 2c). Interestingly, skewing of NK cells to the NKG2C single positive cell subset was accompanied by a shift towards CD56^dim^ NK cells, which furthermore was accompanied by an increase in cells with substantial loss in MFI for CD56 surface expression (Appendix A). Noteworthy, PC3^PSCA^-IL-2-mIL-15d feeder cells preferentially supported the outgrowth of a dominant NKG2A single-positive NK cell fraction from same donors, with minor NKG2C single-positive and NKG2A/NKG2C double-positive subsets with mean frequencies below 10% (Figure 2c).

### 2.2. Expansion by HLA-E*spG-Modified Feeder Cells Results in Increased Frquencies of NK Cells Expressing CD16, KIR2DL2/L3/S2 and CD25

The next experiments focused on surface markers of NK cells linked to tolerance to self (KIRs), maturation (CD57), proliferation (CD25), activating receptors (CD16, NKG2D) and exhaustion (PD-1, LAG-3) as depicted for a representative donor in Figure 3a–c. More specifically, the percentage of NK cells positive for the marker expression was compared between total NK cells expanded with PC3^PSCA^-IL-2-mIL-15d-HLA-E*spG and PC3^PSCA^-IL-2-mIL-15d feeder cells. As control, fresh NK cells from the same donors were included in the analysis. Total NK cells expanded with HLA-E*spG feeder cells showed a shift to increased fractions of CD57+ cells indicating matured NK cells, which was still below the high CD57 frequency found in fresh NK cells (Figure 3d). Interestingly, NK cells expanded by PC3^PSCA^-IL-2-mIL-15d feeder cells barely comprised CD57+ NK cells. No significant differences in the frequency of NK cells expressing KIR3DL1 were detected when comparing fresh NK cells and NK cell samples expanded with the two different feeder cell lines. Frequencies for KIR2DL1/S1/S4+ cells in NK cell populations expanded by both feeder cell lines were highly variable. NK cells expanded by the HLA-E*spG-modified feeder cells exhibited a modest yet statistically non-significant increase in the mean frequency of the KIR2DL1/S1/S4+ subset when compared to NK cells expanded by PC3^PSCA^-IL-2-mIL-15d feeder cells which comprised dominant NKG2A single-positive NK cell populations. However, the mean frequency of KIR2DL1/S1/S4+ subsets was significantly higher in NK cells expanded by the HLA-E*spG-modified feeder cells when compared to fresh NK cells (Figure 3d). Noteworthy, relative KIR2DL2/L3/S2 expression in NK cells expanded by the HLA-G*spG-modified feeder cell line was in the mean 79.2% (±8.1%), which was significantly higher than in fresh NK cells (44.1% ± 10.6%) and NK cells expanded by PC3^PSCA^-IL-2-mIL-15d cells (53.2% ± 12.6%) (Figure 3d). Relative CD25 surface expression was increased in NK cells expanded by PC3^PSCA^-IL-2-mIL-15d-HLA-E*spG feeder cells when compared to fresh NK cells, which barely express CD25, whereas no significant differences were detected when compared to NK cells expanded with PC3^PSCA^-IL-2-mIL-15d feeder cells (Figure 3d). Similar outcomes were obtained for the exhaustion markers PD-1 and LAG-3, which were moderately increased in NK cells expanded by PC3^PSCA^-IL-2-mIL-15d-HLA-E*spG feeder cells (Figure 3d).

A high frequency of cells with expression of CD16, with a mean of 66.8% (±19.4%) was also noted in NK cells expanded by PC3^PSCA^-IL-2-mIL-15d-HLA-E*spG feeder cells which, however, was somewhat lower as compared to frequencies observed in fresh NK cells (83.7% ± 7.8%). In comparison to both, NK cells expanded by PC3^PSCA^-IL-2-mIL-15d feeder cells had a significantly lower frequency of CD16+ NK cells (41.9% ± 11.1%). On the contrary, no significant differences in the frequencies of NKG2D in the differentially expanded NK cells and fresh NK cells were noted (Figure 3d).

A detailed comparison of the NKG2C-positive and NKG2C-negative subsets in the differentially expanded NK cells revealed in all cases a relative and significant increase in CD16 surface frequency in NKG2C-positive subsets when compared to NKG2C-negative subsets (Figure 4). The increase in CD16 frequency in NKG2C-positive NK cells was accompanied by a shift in maturation indicated by an increase in CD57 frequency. Interestingly, NKG2C-positive NK cells expanded with PC3^PSCA^-IL-2-mIL-15d-HLA-E*spG feeder cells, exhibited similar high (>75%) frequencies of CD16 when compared to fresh NK cells. When comparing NKG2C-positive and NKG2C-negative cell fractions in the differentially expanded NK cell populations and in corresponding fresh peripheral NK cells, KIR2DL1/S1/S4 and KIR3DL1 frequencies were highly variable. However, a statistically non-significant increase in KIR2DL1/S1/S4 frequency was linked to NKG2C-positive NK cell fractions when compared to NKG2C-negative cells, irrespective of the mode of expansion. Of note, a high KIR2DL2/L3/S2 frequency >80% was seen in all NKG2C-positive NK cell fractions, which was significantly increased when compared to the corresponding NKG2C-negative NK cell fractions. Interestingly, NKG2C-positive cells expanded with PC3^PSCA^-IL-2-mIL-15d-HLA-E*spG feeder cells displayed an even higher KIR2DL2/L3/S2 frequency than NKG2C-positive NK cells from peripheral blood. Yet, no differences were found between fresh and differentially expanded NKG2C-positive as well as NKG2C-negative NK cell subsets when comparing relative expression levels of NKG2D. In contrast, CD25, the α-chain of the high affinity IL-2 receptor, was detected at similar frequencies in differentially expanded NKG2C-positive and NKG2C-negative NK cells but not in fresh NK cells. Noteworthy, relative PD-1 and LAG-3 expression levels were somewhat increased in the differentially expanded NK cells. In particular, a moderate but significant increase in mean LAG-3 frequency was noted in NKG2C-positive NK cells expanded by HLA-E*spG-modified feeder cells, when compared to the NKG2C-negative NK cell fraction and when compared to NKG2C-positive and -negative NK cell subsets from peripheral blood. This observation together with the observed decreased MFI for CD56 in expanded NKG2C-positive cells might be linked to the continuous stimulation of NK cells with the artificial HLA-E*spG ligand (Appendix A).

### 2.3. Expanded NKG2C+ NK Cells React against K562 Target Cells and Kill Primary Glioblastoma Cells

To assess whether NKG2C+ NK cells are suitable for potential clinical use, in particular for immunotherapy of glioblastoma, we investigated the cytotoxic reaction of expanded NKG2C+ NK cells against K562 cells devoid of HLA class I expression and primary glioblastoma cells with expression of classical HLA molecules.

Because of fluctuations in the expression levels of HLA-E and HLA-G in K562 and GBM cells during cultivation, analysis of HLA surface expression levels was performed simultaneously with the cytotoxicity assays. As expected, we revealed that K562 cells were devoid of HLA class I expression. Furthermore, surface expression of HLA-E and HLA-G was detected on K562 cells (Figure 5a).

Due to the donor-dependent variance in the production of NK cells affecting frequency of NKG2C+ NK cells, magnetic-activating cell sorting (MACS) was employed for enrichment of NKG2C+ NK cells prior to cytotoxicity assays. Of note, positive sorting of cells with anti-NGK2C antibodies was avoided to obviate a recently described receptor-mediated activation and internalization of CD94/NKG2C [22,32]. Instead, we depleted NKG2A+ NK cells and therefore preferentially enriched NKG2C+ NK cells. After sorting, we routinely observed NKG2C+ NK cell purity > 90% (data not shown). Yet, starving of enriched NKG2C+ NK cells overnight prior to the cytotoxicity assay leads to the appearance of NK cell fractions positive for NKG2A and NKG2C (Figure 5b). For chromium release assays, similarly starved NK cells expanded by the PC3^PSCA^-IL-2-mIL-15d feeder cell line, which facilitates outgrowth of the NKG2A+ NK cell subset, and when available fresh NK cells from the same donors, were included for comparison.

By using NK cells from two donors, we analyzed the ability of sorted and expanded NKG2C+ and of expanded NKG2A+ NK cells, respectively, to lyse K562 target cells. Intriguingly, NKG2C+ cells from two donors developed a robust cytotoxic response towards K562 cells, whereas NKG2A+ NK cells show less cytotoxic responses when confronted with target cells (Figure 5c). This results indicate, that HLA-E:NKG2C ligation enhances cytotoxicity of NK cells whereas binding of inhibitory NKG2A to HLA-E restrains NK cell cytotoxicity.

Furthermore, the cytotoxicity of NKG2C+ and of NKG2A+ NK cells against primary GBM cells were tested using the chromium release assay. Flow cytometry analysis of HLA molecules on HT7606 and HT18584 GBM cells revealed strong expression levels for HLA class I molecules, and expression of HLA-E and HLA-G molecules (Figure 6a). HT18816, a primary GBM cell line, additionally included in the study, exhibited diminished HLA-ABC, HLA-E and HLA-G expression when compared to the other primary GBM cell lines (Figure 6a). To detect possible allogeneic reactivity, the HLA genes of the target cells and KIR genes of donors were genotyped. We calculated that HT18584 GBM cells were fully compatible with iKIRs from donors, whereas HT7606 and HT18816 cells show C2/Bw4 and Bw4 mismatches in the graft versus host (GvH) direction, respectively (see inlets in Figure 6b). So far, we observed cytotoxic responses of sorted NKG2C+ NK cells against primary GBM target cells whereas NKG2A+ NK cells did not respond. In addition, fresh NK cells from the same donors could be included in some cytotoxicity assays but did not react against the primary tumor cells (Figure 6b). Of note, when we probed NKG2C+ NK cells against HT7606 lacking dominant C1 and devoid of the weaker Bw4 ligand for iKIRs, we revealed very strong cytotoxic reactions. However, such a high NKG2C+ NK cell cytotoxicity was not observed against HT18816 target cells, having a Bw4 mismatch and HT18584 target cells expressing all cognate HLA-ligands for donor iKIRs. This might indicate a decreased threshold for NK cell activation when HLA-C ligands for inhibitory KIRs are missing, which could enhance cytotoxic response of NKG2C+ NK cells. Interestingly, NK cells from the donors contained the genes 2DS2 and 2DS4 encoding activating KIRs. Whereas no ligand for KIR2DS2 were found on all tumor cells, C*05:01 and C*02:02 alleles serving as ligands for KIR2DS4 [33] were present in HT18584 and HT18816 primary GBM cells, respectively. Theoretically, a potential alloreactivity of sorted NKG2C+ NK cells should, to some extent, be paralleled in expanded NKG2A+ NK cells from the same donor also positive for KIR2DL1/S1/S4 surface expression. However, NKG2A+ NK cells, and fresh NK cells, irrespective of the presence of KIR2DS4 alleles and missing ligands for iKIRs, did not react towards HLA-E+/HLA-G+ primary GBM cells (Figure 6b).

To further confirm that ligation of NKG2C with HLA-E can overcome inhibitory KIR-signaling the primary GBM cell line HT18584 was genetically modified with HLA-E*spG to generate HT18584-HLA-E*spG cells. After antibiotic selection, the expression of the HLA-E*spG transgene was confirmed by HLA-E surface staining which demonstrated a 26-fold increased MFI for HLA-E-signals when compared to parental HT18584 cells (Figure 7a). Sorted NKG2C+ NK cells from two donors showed an enhanced cytotoxicity against HT18584-HLA-E*spG cells when compared to cytotoxicity towards parental HT18584 cells (Figure 7b). In summary, the data indicate that activation through NKG2C is involved in the subtle tuning of NK cell reactivity towards glioblastoma cells expressing HLA-E and HLA-G.

## 3. Discussion

The unique capability of NK cells to distinguish normal from transformed cells, and their rapid and efficient killing capacity, have led to them becoming an attractive option for immunotherapy. In particular, in glioblastoma the expression of both HLA-E and HLA-G might be exploited for an immunotherapy with adaptive NKG2C+ NK cells, which specifically recognize HLA-E molecules presenting a high affinity peptide derived from HLA-G.

So far, NKG2C+ cells generally have been considered to unfold a higher antibody-dependent cellular cytotoxicity (ADCC) and subsequent cytokine release when compared to conventional NK cells yet are supposed to have weak natural effector function. This functional difference of NKG2C+ NK cells might be related to (i) increased CD16/FcγRIII expression, loss of FcεRIγ signaling adapter and favored signaling via CD3ζ and (ii) elevated inhibitory KIR expression inducing tolerance to tumor cells exhibiting protective amounts of HLA class I molecules [22,34,35]. Interestingly, we report a shifting towards CD56^dim^ and relatively high expression levels of CD16 in NKG2C+ NK cells when compared to NKG2A+ NK cells, which was similar to fresh NK cells, and together with the observed increased frequencies of CD57 expression, indicate a shift to a terminally differentiated effector phenotype during cultivation on HLA-E*spG-modified feeder cells [36]. In line with previous findings, we also confirmed increased frequency of KIR2DL2/L3/S2 expression, in NK cells expanded with PC3^PSCA^-IL-2-mIL-15d-HLA-E*spG, in particular in the NKG2C+ NK cell subset when compared to fresh NK cells and NKG2A+ NK cells [34]. Noteworthy, a significant difference in mean relative expression levels of KIR2DL1/S1/S4 and KIR3DL1 was not observed when comparing NKG2C+ and NKG2A+ NK cells. We furthermore noticed a higher frequency of CD25 expression in expanded cells, and in particular, in selectively expanded NKG2C+ cells we detected a significantly increased fraction of NK cells with strongly decreased MFI for CD56 and a moderately increased frequency for the exhaustion marker LAG-3. This might be due to enduring stimulation of NK cells via the HLA-E*spG artificial ligand on PC3^PSCA^-IL-2-mIL-15d-HLA-E*spG feeder cells. Likewise, pronounced loss of CD56 density on NK cells was also detected in chronic HIV [37] and HCV [38] infections, as well as in healthy patients with latent co-infection of HCMV and EBV [39]. Intriguingly, such chronically stimulated NK cells were reported to exhibit decreased effector functions. Incidentally, a recent study demonstrated epigenetic reprogramming and dysfunctional NK cells after chronic stimulation through surface coated NKG2C antibodies [40]. In contrast to the aforementioned studies, the majority of our expanded NKG2C+ NK cells remained functionally competent in killing tumor cells as discussed below. In this regard, it is conceivable that our expansion protocol, in particular the low feeder cell to NK cell ratio of 1:40, might attenuate chronic stimulation via NKG2C and eventually results in higher rates of functional NKG2C+ NK cells.

Concerning natural cytotoxicity of NKG2C+ NK cells towards tumor cells, several reports indicated efficient lysis of K562 and LCL 722.221 target cells devoid of HLA molecules by NKG2C+ NK cells, which was indistinguishable or higher when compared to the lytic efficiency of conventional NKG2C-negative NK cells [22,41]. In our study, we observed an increased cytotoxic response of NKG2C+ NK cells from two donors when confronted with K562 target cells as compared to NKG2A+ NK cells. This result is in line with previous reports demonstrating increased cytotoxicity of NKG2C+ NK cells when encountering 722.221-AEH cells, known to express HLA-E loaded with a VMAPRTLVL peptide [41,42].

In our study, we corroborated for the first time, a natural and specific cytotoxicity of NKG2C+ NK cells towards HLA class I-positive glioblastoma cells expressing HLA-E and HLA-G. Although the VMAPRTLFL peptide derived from HLA-G in GBM cells has highest affinity for CD94/NKG2C reported so far [23], it is conceivable that competing peptides, such as those derived from classical HLA, can also be presented via HLA-E and therefore can modulate the cytotoxic response of NKG2C+ NK cells. In addition, differential expression of stress molecules (i.e., MICA, ULBPs) on GBM cells, serving as ligand for NKG2D might have modulated activation and cytotoxic responses of NKG2C+ NK cells against GBM cells [43].

So far, the KIR genotype of donors, in particular missing ligands for iKIRs, might also have contributed to the cytotoxic response of NKG2C+ cells to primary GBM. Concerning aKIRs, a KIR2DS4 allele and increased NK cell subpopulations expressing CD16, NKG2D, and CD94/NKG2C were recently described to prolong survival of GBM patients [44]. In our study, we cannot completely rule out that KIR2DS4 influenced subtle tuning of NKG2C+ NK cell cytotoxicity towards HT18584 and HT18816 GBM target cells with expression of the cognate HLA ligands. However, expanded NKG2A+ NK cells from the same donors having the KIR2DS4 allele failed to develop a cytotoxic response towards HT18584 and HT18816 GBM target cells with expression of cognate C*02:02 and C*05:01 ligands, respectively, which might indicate the absence of aKIR-mediated alloreactivity in both differentially expanded NK cells. That HLA-E:NKG2C ligation indeed may play a role in enhanced cytotoxicity of NK, even in an HLA:KIR match scenario, was demonstrated by a significantly increased cytotoxic response towards allogeneic GBM cells engineered to express HLA-E*spG.

In conclusion, the results indicate potential therapeutic value of NKG2C+ NK cells for treatment of gliomas and of tumors expressing adequate amounts of HLA-E. Further work and in vivo studies are warranted to improve expansion of NK cells and simultaneously avoiding their exhaustion to confirm feasibility of NKG2C+ NK cells for clinical studies.

## 4. Materials and Methods

### 4.1. Cell Lines

The NK cell feeder cell line PC3^PSCA^-IL-2-mIL-15d has been described recently [31]. PC3^PSCA^-IL-2-mIL-15d cells, the novel feeder cell line PC3^PSCA^-IL-2-mIL-15d-HLA-E*spG, and the erythroleukemic K562 cell line, were maintained in RPMI-1640 medium (Life Technologies, Carlsbad, CA, USA) with 10% *v*/*v* heat-inactivated fetal bovine serum (Life Technologies), 2 mM L-glutamine (Life Technologies), 10 mM HEPES (Life Technologies), 100 U/mL penicillin (Life Technologies) and 0.1 mg/mL streptomycin (Life Technologies). The human embryonic kidney cell line HEK293T was used for production of lentiviral particles and maintained in DMEM containing 4.5 g/L glucose (Life Technologies) supplemented with 10% *v*/*v* heat-inactivated FBS (Life Technologies), 10 mM HEPES (Life Technologies), 100 U/mL penicillin (Life Technologies), and 0.1 mg/mL streptomycin (Life Technologies). The local ethical committee of the Medical Faculty Carl Gustav Carus, TU Dresden (#EK 323122008, 22 January 2009), approved the use of primary GBM cells from patients. The HT7606 GBM cell line has been already described [45] and the primary GBM cell lines HT18584 and HT18816 were prepared by using the Brain Tumor Dissociation Kit (P) (Miltenyi Biotec, Bergisch Gladbach, Germany) and afterwards cultivated in fully complemented DMEM as described above. All cell lines were authenticated (Multiplexion GmbH, Heidelberg, Germany), and cultivated at 37 °C, 5% CO_2_ in 80% humidified environment.

### 4.2. Lentiviral Vectors Encoding HLA-E*spG and Transduction of Cells

Lentiviral particles were produced by a transient three vector packaging protocol [25], and transductions were performed as described previously [31]. The self-inactivating lentiviral pHATtrick-Hygro^R^ vector [25] modified with a previously described disulfide-stabilized HLA-E trimer consisting of β2-microglobulin, VMAPRTLFL-peptide and HLA-E*01:03 ectodomain [22], which was complemented with the transmembrane and cytoplasmic domain of HLA-E, was used to transduce PC3^PSCA^-IL-2-mIL-15d feeder cells and HT18584 primary GBM cells. Transduced cells were selected with 200 µg/mL hygromycin B (Life Technologies) for two weeks.

### 4.3. Isolation and Expansion NK Cells

The local ethical committee of the Medical Faculty Carl Gustav Carus, TU Dresden (#EK 242102007, 2 November 2007), approved the use of human PBMCs including NK cells from healthy donors. PBMCs were isolated by Biocoll gradient centrifugation (Biochrom, Berlin, Germany) and untouched NK cells were prepared using the negative NK Cell Isolation Kit (Miltenyi Biotec). Staining with anti-CD56-APC (clone REA196) and anti-CD3-PE (clone REA613) antibodies (Miltenyi Biotec) routinely confirmed ≥85% purity of CD56+ NK cells and depletion of CD3+ T cells and NKT cells (Appendix A). Freshly isolated NK cells were maintained in NK MACS complete-medium consisting of NK MACS medium (Miltenyi Biotec) supplemented with 1% NK MACS supplement (Miltenyi Biotec) and 5% human AB serum (c.c.pro Gesellschaft für Herstellung und Vertrieb von Produkten für Cellculturen mbH, Oberdorla, Germany). 24 h before expansion 2.5 × 10^4^ feeder cells were seeded in a 24-well plate with 1 mL complete RPMI-1640 medium. Afterwards, the medium was replaced with 1 mL NK MACS complete-medium for 4 h to allow conditioning of medium by the feeder cells. Then 1 × 10^6^ NK cells in 1 mL complete NK MACS medium were added to the feeder cells to start expansion. NK cells were passaged at day 3, 7, and 11 onto fresh feeder cells and harvested at day 14 for experiments. Enrichment of NKG2C-positive NK cells was accomplished by washing NK cells with pre-warmed medium at least twice before applying negative MACS using anti-NKG2A-biotin antibodies according to the instructions of the supplier (Miltenyi Biotec, Bergisch Gladbach, Germany).

### 4.4. HLA and KIR Genotyping

HLA and KIR genotyping were performed as described previously [31]. IPD-KIR Ligand Calculator (EMBL-EBI, Hinxton, UK) was employed to identify KIR:KIR-ligand mismatches in the graft versus host direction when NK cells were confronted with target cells.

### 4.5. Flow Cytometry Analyses of Cell Surface Proteins

The purity of NK cells was routinely tested by fluorescence flow cytometry of 2 × 10^5^ cells stained with anti-CD56-APC (clone REA196) and anti-CD3-PE (clone REA613) (Miltenyi Biotec). Monitoring of selective expansion of the NKG2C+ NK cell subset was performed using staining with anti-CD56-APC, anti-NKG2C-PE (clone REA205) and anti-NKG2A-FITC (clone REA110) (Miltenyi Biotec). For phenotypic analysis of the NKG2C+ NK cell subset 2 × 10^5^ cells were simultaneously stained with anti-NKG2C-APC (clone REA205) and anti-NKG2A-FITC (clone REA110) (Miltenyi Biotec) in combination with one of the following fluorochrome-coupled antibodies: anti-CD314(NKG2D)-PE (clone REA1175), anti-CD25-APC (clone REA570), anti-CD279(PD-1)-PE (clone REA802), anti-CD57-PE (clone REA769), anti-CD16-VioBlue (clone REA423), anti-CD223(LAG-3)-VioBlue (clone REA351), anti-KIR2DL2/L3/S2-PE (clone DX27) and anti-KIR3DL1-PE (clone DX9) (Miltenyi Biotec) or anti-KIR2DL1/S1/S4-PE (clone HP-3E4, BD Pharmingen, USA). Endogenous expression of classical and non-classical HLA on cells were assessed using anti-HLA-ABC-VioBlue (clone REA230), HLA-E-PE (clone REA1031), HLA-G-APC (clone 87G) (Miltenyi Biotec). Expression of transgenic HLA-E*spG was confirmed by HLA-E-PE staining. Appropriate isotype-antibody controls were included in all experiments. Stained cells were measured using MACSQuant Analyzer 10 flow cytometer (Miltenyi Biotec) and analyzed by FlowJo software version 10.7 (FlowJo, LLC., Ashland, OR, USA).

### 4.6. Chromium (^51^Cr) Release Cytotoxicity Assay

Cytotoxicity of effector cells towards target cells was assessed by chromium release. Briefly, NK cells were starved overnight in complete NK MACS medium devoid of cytokines. Target cells were labelled with ^51^Cr by incubation of 2 × 10^6^ cells in 1.5 MBq sodium ^51^chromate (Hartmann Analytic, Braunschweig, Germany) at 37 °C and 5% CO_2_ for 1 h. Labelled cells were washed at least three times with PBS before seeding 2 × 10^3^ cells per well in triplicates in a 96 round-bottom well plate. Overnight starved NK cells were suspended in fresh complete NK-MACS medium and added to the wells at target to effector ratios of 1:5, 1:7.5 and 1:10. After 20 h the cell culture supernatant (25 µL per well) was added to 150 µL of scintillation solution Ultima Gold (PerkinElmer, Waltham, MA, USA) in a 96 well plate and well mixed by plate shaking. Wallace 1450 Microbeta Trilux Liquid Scintillation and Luminescence Counter (PerkinElmer, Waltham, MA, USA) was used to measure released chromium. Maximal release was measured by treating target cells with 5% Triton X-100 (Serva, Heidelberg, Germany) and minimum release by cultivation of target cells with medium alone. Percentage of specific lysis was calculated using the standard formula: 100 × (cpm release target cells − cpm minimum release)/(cpm maximum release − cpm minimum release).

### 4.7. Statistical Analysis

For comparing results of surface marker expression and of chrome release assays when using differentially expanded and fresh NK cells, respectively, one-way ANOVA followed by Tukey’s multiple comparison test was used or two-way ANOVA followed by Tukey’s multiple comparison test when considering two independent factors. To compare loss of MFI of CD56 in NK cells expanded by two different feeder cell lines a paired t test (Wilcoxon matched-pairs signed rank test) for matched samples was used. For assessing statistical differences in relative NKG2C expression in NK cells of HCMV-seropositive and naïve NK cell donors an unpaired t test with Welch’s correction for unmatched samples was employed. Statistical data in the figures represented as mean ± SD, *p* value less than 0.05 indicates significant difference. GraphPad Prism Software version 9.3.0 (Graphpad Software, Inc., San Diego, CA, USA) used to perform statistical analysis.

## Figures and Tables

**Figure 1 ijms-23-05857-f001:**
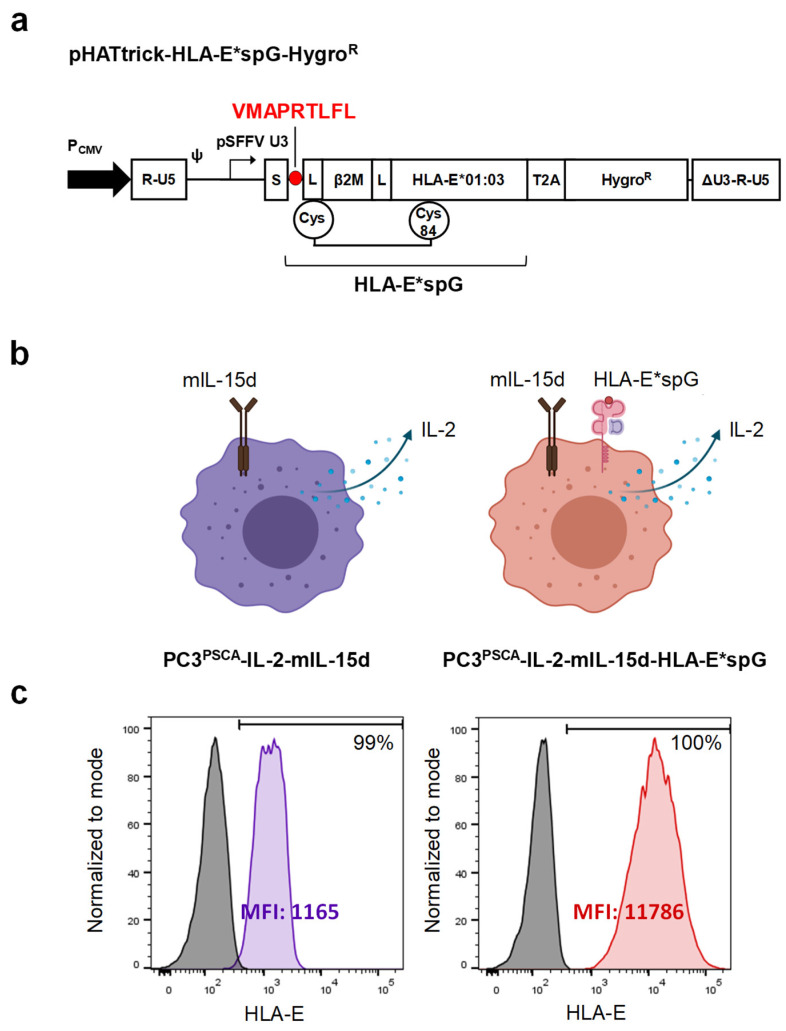
Generation of PC3^PSCA^-IL-2-mIL-15d-HLA-E*spG feeder cells. (**a**) Schematic representation of lentiviral construct encoding artificial disulfide-stabilized single chain trimer (dSCT) consisting of β2-microglobulin and the HLA-G signal peptide derived nonamer fused to leaderless HLA-E coding sequence and 3′-downstream T2A-hygromycin resistance gene. S depicts an influenza virus hemagglutinin H1N1 leader sequence; L depicts glycine-serine linkers. Note that the first linker contains a cysteine, which forms a disulfide bridge with a cysteine substituting tyrosine at position 84 of the α1-domain of HLA-E*01:03. Expression of the integrated provirus is driven by a spleen focus forming virus U3 promoter/enhancer region (pSFFV U3). (**b**) Left: Schematic drawing of the parental PC3^PSCA^-IL-2-mIL-15d feeder cell expressing a secreted form of IL-2 and a membrane-bound IL-15 (mIL-15d) consisting of the coding sequence of human IL-15 fused to mutant DAP12 devoid of functional ITAMs. Right: Schematic drawing of PC3^PSCA^-IL-2-mIL-15d-HLA-E*spG feeder cell with additional expression of dSCT for HLA-E. (**c**) Flow cytometry analysis of PC3^PSCA^-IL-2-mIL-15d (purple histogram) and PC3^PSCA^-IL-2-mIL-15d-HLA-E*spG feeder cells (red histogram) stained with antibody specific for HLA-E. Isotype staining controls are included (grey histograms).

**Figure 2 ijms-23-05857-f002:**
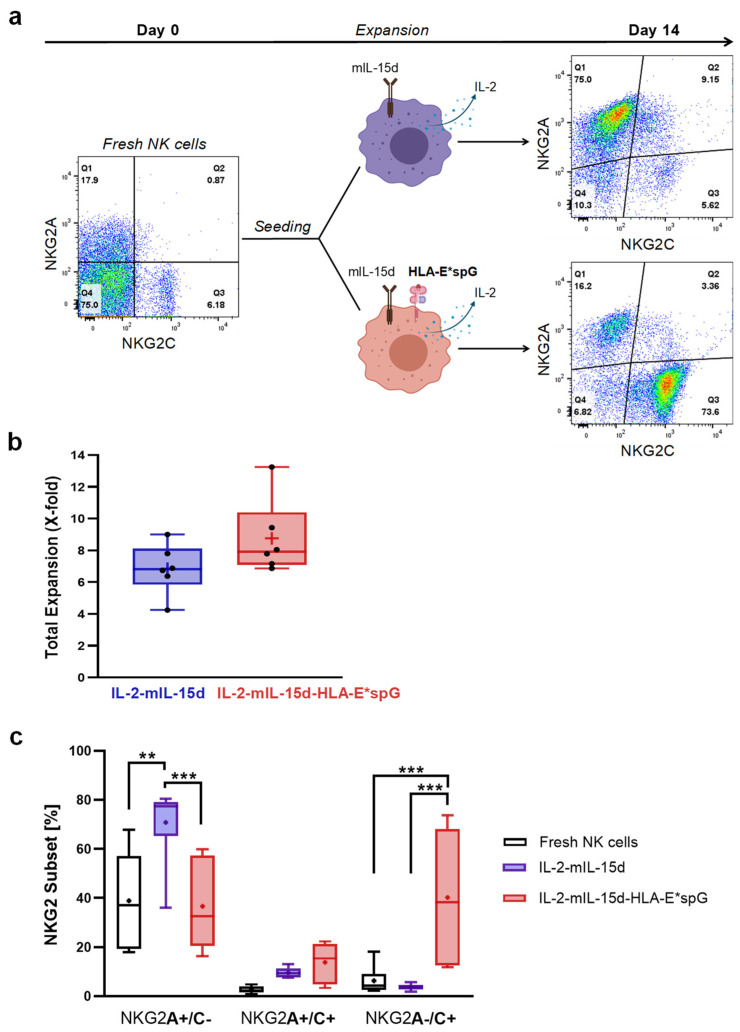
Selective expansion of NKG2C and NKG2A NK cell subsets with feeder cells. (**a**) Flow cytometry analysis of NKG2A and NKG2C expression at day 0 in CD56+/CD3− NK cells from one representative donor followed by expansion using the depicted feeder cell lines and analysis of NKG2A and NGK2C NK cell subsets at day 14. (**b**) After 14 days of expansion, NK cells were counted, and the expansion factor (x-fold of starting number of NK cells) was calculated (*n* = 6). (**c**) Analysis of NKG2+ NK cell subsets in the differentially expanded NK cells and in corresponding fresh NK cells from same donors (*n* = 6). PC3^PSCA^-IL-2-mIL-15d-HLA-E*spG feeder cells promote selective outgrowth and therefore a higher frequency of NKG2C single-positive NK cells when compared to fresh NK cells and NK cells expanded with PC3^PSCA^-IL-2-mIL-15d feeder cells. Vice versa, PC3^PSCA^-IL-2-mIL-15d feeder cells preferentially promote the generation of NKG2A single-positive NK cells. Dots and the horizontal lines inside the Whisker boxes indicate the mean and the median. ** *p* < 0.01, *** *p* < 0.001.

**Figure 3 ijms-23-05857-f003:**
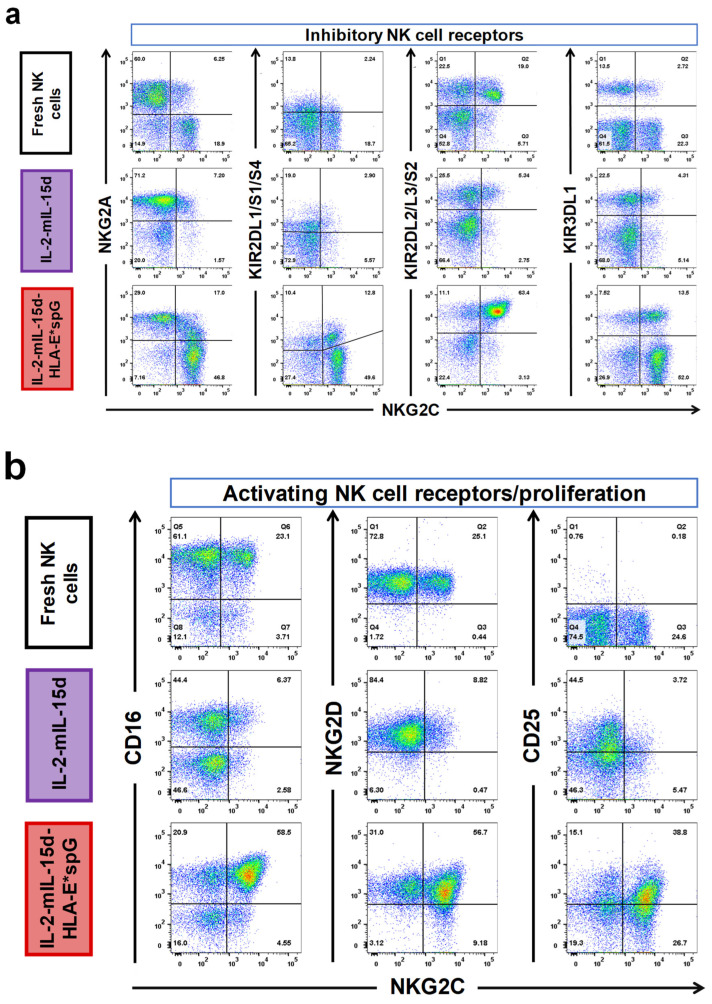
Phenotypic analysis of differentially expanded NK cells and fresh NK cells. Depicted is a representative flow cytometry analysis from expanded and fresh NK cells from one donor. As described in the material and methods section, NK cells were stained at day 14 after the start of expansion with an NKG2C-specific antibody and were simultaneously stained with indicated antibodies (**a**) specific for inhibitory receptors NKG2A, KIR2DL1/S1/S4, KIR2DL2/L3/S2 and KIR3DL1, (**b**) specific for activating receptors CD16, NKG2D and for the activation/proliferation marker CD25 as well as (**c**) specific for surface markers related to exhaustion/immune checkpoint and maturation (PD-1, LAG-3, CD57). (**d**) Quantitative analysis of marker expression in differentially expanded total NK cells and total fresh NK cells. For analysis of KIRs, CD16, NKG2D, CD25 and PD-1 six expansion experiments from six donors are included. For analysis of CD57 and LAG-3 four expansion experiments from three donors are included. Dots and the horizontal lines inside the Whisker boxes indicate the mean and the median. * *p* < 0.05, ** *p* < 0.01, *** *p* < 0.001.

**Figure 4 ijms-23-05857-f004:**
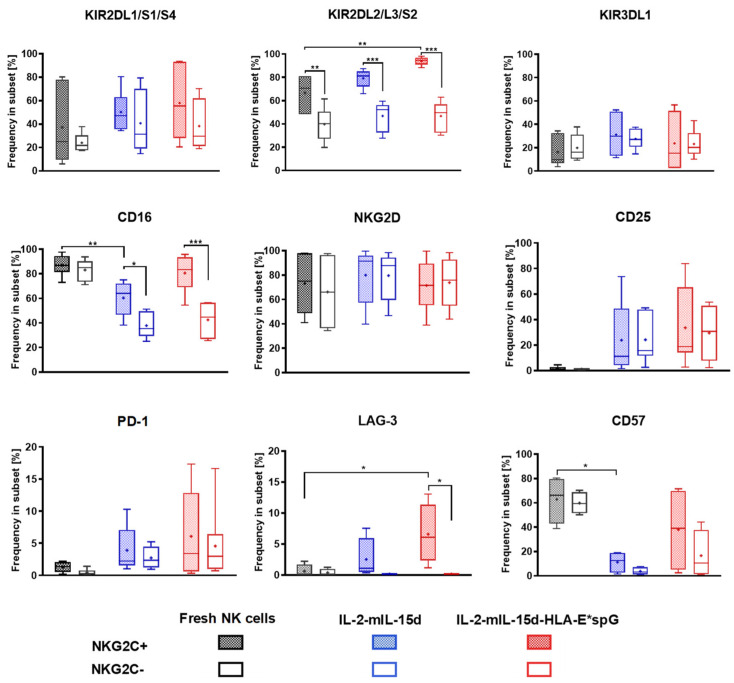
Relative expression of surface markers in NKG2C-positive and NKG2C-negative NK cells after expansion with different feeder cells. Depicted is the quantitative analysis of relative marker expression in gated NKG2C-positive and NKG2C-negative CD56+ NK cells in differentially expanded NK cells, respectively. Fresh NK cells from the same donors were included in the experiments. For calculation of relative marker expression, gated NKG2C-positive and NKG2C-negative NK cell fractions were both set as 100% and percentage of markers of corresponding NK cell fractions were calculated. For analysis of KIRs, CD16, NKG2D, CD25 and PD-1 six expansion experiments from six donors are included. For analysis of LAG-3 and CD57 four expansion experiments from three donors are included. Dots and the horizontal lines inside the Whisker boxes indicate the mean and the median, respectively. * *p* < 0.05, ** *p* < 0.01, *** *p* < 0.001.

**Figure 5 ijms-23-05857-f005:**
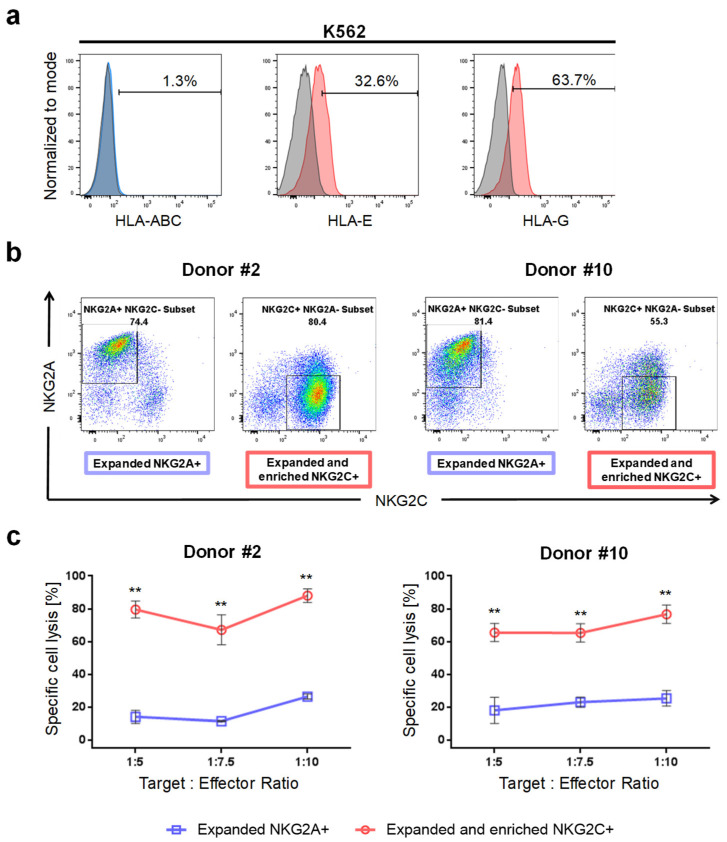
Cytotoxicity of NKG2C+ and NKG2A+ NK cells towards K562 target cells. (**a**) Flow cyto-metry analysis of HLA class I, HLA-E and HLA-G surface expression in K562 target cells. K562 cells are devoid of classical HLA but express HLA-E and HLA-G. Isotype staining controls are included (grey histograms) (**b**) Flow cytometry analysis of expanded NKG2A+ NK cells and of expanded and successively sorted NKG2C+ NK cells from donors #2 and #10 showing purity of the NK cell subsets prior to the chromium release assay. (**c**) Chromium release assay demonstrating preferential killing of K562 target cells by NKG2C+ NK cells from two donors at indicated target to effector ratios when compared to NKG2A+ NK cells and fresh NK cells. ** *p* < 0.01.

**Figure 6 ijms-23-05857-f006:**
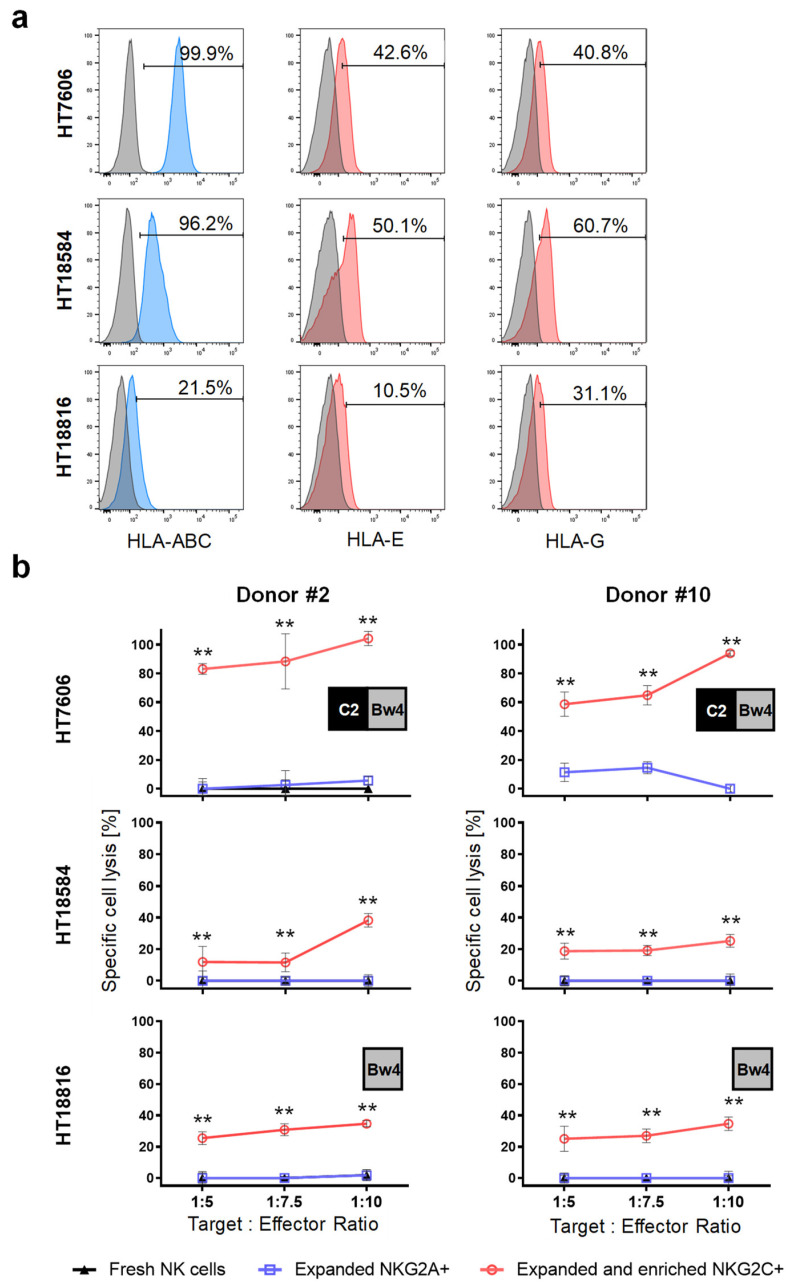
Cytotoxicity of NKG2C+ NK cells towards primary HLA-E+/HLA-G+ glioblastoma cells. (**a**) Flow cytometry analysis of HLA class I as well as of HLA-E and HLA-G surface expression in HT7606, HT18584 and HT18816 primary glioblastoma cells. As indicated, all GBM cells express HLA class I molecules and are positive for HLA-E and HLA-G. Isotype staining controls are included (grey histograms). (**b**) Chromium release assay demonstrating preferential killing of HLA class I-positive glioblastoma cells by NKG2C+ NK cells from two donors at indicated target to effector ratios when compared to NKG2A+ NK cells and fresh NK cells. Potential alloreactivity (i.e., missing HLA self-ligand for KIR) is depicted in inlets. ** *p* < 0.01.

**Figure 7 ijms-23-05857-f007:**
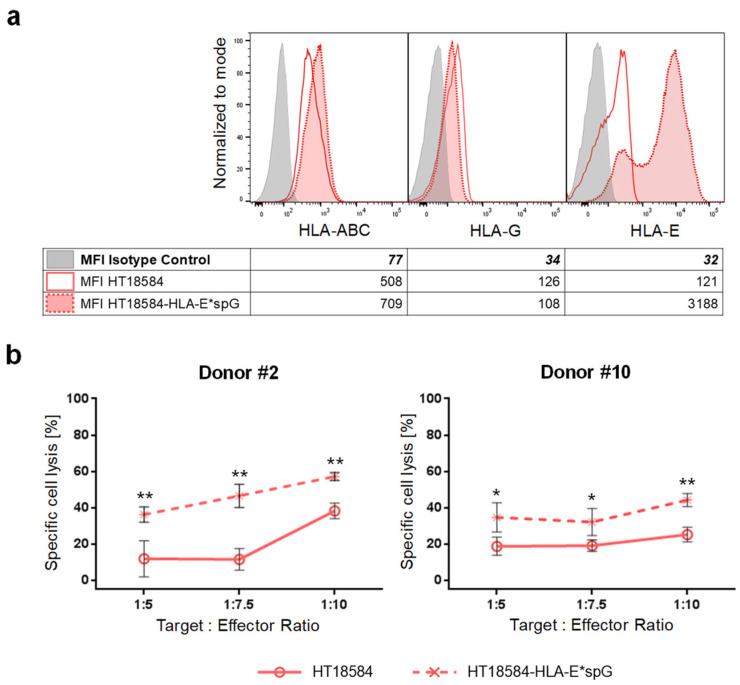
Ectopic expression of HLA-E*spG significantly increases susceptibility of HLA/KIR-matched GBM target cells towards NKG2C+ NK cell-mediated lysis. (**a**) Flow cytometry analysis of HLA class I as well as of HLA-G, and HLA-E/HLA-E*spG surface expression in parental HT18584 and HT18584-HLA-E*spG glioblastoma cells. Isotype staining controls are included (grey histograms). As depicted in the accompanying table, the transgenic expression of HLA-E*spG results in an increased MFI for HLA-E but expression levels for HLA class I and HLA-G remain mostly unchanged. (**b**) Chromium release assay demonstrating increased killing of HT18584-HLA-E*spG target cells by sorted NKG2C+ NK cells when compared to parental HT18584 cells. * *p* < 0.05, ** *p* < 0.01.

## Data Availability

The data presented in this study are available from the corresponding author on request.

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
