# Peer review of "NKG2C+ NK Cells for Immunotherapy of Glioblastoma Multiforme"

_ijms, 2022, doi:10.3390/ijms23105857_

Round 1
Reviewer 1 Report
In this well-written paper authors explore the possibility to use NKG2C+ expanded NK cells as adoptive transfer therapy of GBM patients. Methods are well described and all experiments were performed using the appropriate controls. The use of patient-derived primary cultures further strengthen the conclusions.
I'm only surprised that authors performed cytotoxicity assays using 20h co-cultures. Have the author the chance to test NKG2C+ cells ability to kill GBM cells a classical 4h Cr cytotoxic assay? I'm wondering whether also in this case NKG2C+ cells show a higher cytotoxic ability compared to NKG2A+ cells.
Author Response
We thank Reviewer 1 for his comments and would like to answer the questions in detail:
Question 1: Have the author the chance to test NKG2C+ cells ability to kill GBM cells a classical 4h Cr cytotoxic assay?
Unfortunately, we do not have data on the cytotoxicity of NKG2C+ NK cells after 4 h, because in our group the standard chromium release assay is performed for 18-21 h. We have already published this in other publications like this (Müller et al., 2015, doi: 10.1097/CJI.0000000000000082; Töpfer et al., 2015, doi: 10.4049/jimmunol.1400330; Michen et al., 2020, doi: 10.1016/j.jcyt.2020.02.004)
Question 2: I'm wondering whether also in this case NKG2C+ cells show a higher cytotoxic ability compared to NKG2A+ cells.
Since we do not have corresponding data after 4h, we can only speculate that this is already the case after 4h. This is supported by the fact that the cytotoxicities of the NKG2A+ NK cells are below 20% in almost all experiments (Figs. 5 and 6), even after 20h. In future experiments we can investigate this.
Reviewer 2 Report
This manuscripts by Murad et al presents the production of novel feeder cells for the expansion of NKG2C+ NK cells. The authors build on their previous work and add an artificial disulfide-stabilized trimeric HLA-E ligand (HLA-E*spG) to the PC3 cells already expressing IL2, 4-1BBL, and a membrane-bound form of IL-15. Aim is to achieve preferential expansion of NKG2C+ NK cells that are expected to be more cytotoxic against glioblastoma cells. Rationale is to reproduce the preferential expansion observed in samples from HCMV seropositive donors.
The authors perform a precise characterization of the expanded NK cells and confirm that the new feeder cells are able to favor expansion of NKG2C+ NK cells, although not always to the desired extent. The so-expanded NK cells from two donors were active against K562 cells as model for HLA-E/G positive, HLA class I negative cells.
A robust activity could be observed against one of three primary glioblastoma samples. Overexpression of HLA-E*spG in GBM cells could increase cytotoxic activity of NKG2C+ NK cells.
All the data presented in the paper is of good quality, and the paper is very well written. It is clear that the results presented in this work are just one step in the right direction and that a long way has to be covered to reach the final goal of producing NK cells in sufficient amounts, with sufficient specific activity for the therapy of GBM. Nevertheless, the results are encouraging and of general interest to the scientific community.
Author Response
We are very grateful to Reviewer 2 for his/her positive feedback.